# A Bayesian Perspective on Generalization and Stochastic Gradient Descent

**Samuel L. Smith**[*] **& Quoc V. Le**
Google Brain
{slsmith, qvl}@google.com

## Abstract

We consider two questions at the heart of machine learning; how can we predict if a minimum will generalize to the test set, and why does stochastic gradient descent find minima that generalize well? Our work responds to Zhang et al. (2016), who showed deep neural networks can easily memorize randomly labeled training data, despite generalizing well on real labels of the same inputs. We show that the same phenomenon occurs in small linear models. These observations are explained by the Bayesian evidence, which penalizes sharp minima but is invariant to model parameterization. We also demonstrate that, when one holds the learning rate fixed, there is an optimum batch size which maximizes the test set accuracy. We propose that the noise introduced by small mini-batches drives the parameters towards minima whose evidence is large. Interpreting stochastic gradient descent as a stochastic differential equation, we identify the "noise scale" $g = \epsilon(\frac{N}{B} - 1) \approx \epsilon N/B$, where $\epsilon$ is the learning rate, $N$ the training set size and $B$ the batch size. Consequently the optimum batch size is proportional to both the learning rate and the size of the training set, $B_{opt} \propto \epsilon N$. We verify these predictions empirically.

## 1 Introduction

This paper shows Bayesian principles can explain many recent observations in the deep learning literature, while also discovering practical new insights. Zhang et al. (2016) trained deep convolutional networks on ImageNet and CIFAR10, achieving excellent accuracy on both training and test sets. They then took the same input images, but randomized the labels, and found that while their networks were now unable to generalize to the test set, they still memorized the training labels. They claimed these results contradict learning theory, although this claim is disputed (Kawaguchi et al., 2017; Dziugaite & Roy, 2017). Nonetheless, their results beg the question; if our models can assign arbitrary labels to the training set, why do they work so well in practice? Meanwhile Keskar et al. (2016) observed that if we hold the learning rate fixed and increase the batch size, the test accuracy usually falls. This striking result shows improving our estimate of the full-batch gradient can harm performance. Goyal et al. (2017) observed a linear scaling rule between batch size and learning rate in a deep ResNet, while Hoffer et al. (2017) proposed a square root rule on theoretical grounds.

Many authors have suggested "broad minima" whose curvature is small may generalize better than "sharp minima" whose curvature is large (Chaudhari et al., 2016; Hochreiter & Schmidhuber, 1997). Indeed, Dziugaite & Roy (2017) argued the results of Zhang et al. (2016) can be understood using "nonvacuous" PAC-Bayes generalization bounds which penalize sharp minima, while Keskar et al. (2016) showed stochastic gradient descent (SGD) finds wider minima as the batch size is reduced. However Dinh et al. (2017) challenged this interpretation, by arguing that the curvature of a minimum can be arbitrarily increased by changing the model parameterization. In this work we show:

- The results of Zhang et al. (2016) are not unique to deep learning; we observe the same phenomenon in a small "over-parameterized" linear model. We demonstrate that this phenomenon is straightforwardly understood by evaluating the Bayesian evidence in favor of each model, which penalizes sharp minima but is invariant to the model parameterization.

---

[*]Work done as a member of the Google Brain Residency Program (g.co/brainresidency)

- SGD integrates a stochastic differential equation whose "noise scale" $g \approx \epsilon N/B$, where $\epsilon$ is the learning rate, $N$ training set size and $B$ batch size. Noise drives SGD away from sharp minima, and therefore there is an optimal batch size which maximizes the test set accuracy. This optimal batch size is proportional to the learning rate and training set size[1].

We describe Bayesian model comparison in section 2. In section 3 we replicate the observations of Zhang et al. (2016) in a linear model, and show they are explained by the Bayesian evidence. In section 4 we show there is an optimum batch size which maximizes the test set accuracy, and in section 5 we derive scaling rules between the optimum batch size, learning rate, training set size and momentum coefficient. Throughout this work, "generalization gap" refers to the gap in test accuracy between small and large batch SGD training, not the gap in accuracy between training and test sets.

## 2 BAYESIAN MODEL COMPARISON

Bayesian model comparison was first applied to neural networks in MacKay (1992). We provide a brief tutorial here, since the theory is central to the remainder of the paper. For simplicity we first consider a classification model $M$ with a single parameter $\omega$, training inputs $x$ and training labels $y$. We can infer a posterior probability distribution over the parameter by applying Bayes theorem,

$$P(\omega|y, x; M) \quad = \quad \frac{P(y|\omega, x; M)P(\omega; M)}{P(y|x; M)} \tag{1}$$

The likelihood, $P(y|\omega, x; M) = \prod_i P(y_i|\omega, x_i; M) = e^{-H(\omega; M)}$, where $H(\omega; M) = -\sum_i \ln\left(P(y_i|\omega, x_i; M)\right)$ denotes the cross-entropy of unique categorical labels. We typically use a Gaussian prior, $P(\omega; M) = \sqrt{\lambda/2\pi}e^{-\lambda\omega^2/2}$, and therefore the posterior probability density of the parameter given the training data, $P(\omega|y, x; M) \propto \sqrt{\lambda/2\pi}e^{-C(\omega; M)}$, where $C(\omega; M) = H(\omega; M) + \lambda\omega^2/2$ denotes the L2 regularized cross entropy, or "cost function", and $\lambda$ is the regularization coefficient. The value $\omega_0$ which minimizes the cost function lies at the maximum of this posterior. To predict an unknown label $y_t$ of a new input $x_t$, we should compute the integral,

$$P(y_t|x_t, y, x; M) \quad = \quad \int d\omega \, P(y_t|\omega, x_t; M)P(\omega|y, x; M) \tag{2}$$

$$= \quad \frac{\int d\omega \, P(y_t|\omega, x_t; M)e^{-C(\omega; M)}}{\int d\omega \, e^{-C(\omega; M)}}. \tag{3}$$

However these integrals are dominated by the region near $\omega_0$, and since $P(y_t|\omega, x_t; M)$ is smooth we usually approximate $P(y_t|x_t, x, y; M) \approx P(y_t|\omega_0, x_t; M)$. Having minimized $C(\omega; M)$ to find $\omega_0$, we now wish to compare two different models and select the best one. The probability ratio,

$$\frac{P(M_1|y, x)}{P(M_2|y, x)} = \frac{P(y|x; M_1)}{P(y|x; M_2)} \frac{P(M_1)}{P(M_2)}. \tag{4}$$

The second factor on the right is the prior ratio, which describes which model is most plausible. To avoid unnecessary subjectivity, we usually set this to 1. Meanwhile the first factor on the right is the evidence ratio, which controls how much the training data changes our prior beliefs. Germain et al. (2016) showed that maximizing the evidence (or "marginal likelihood") minimizes a PAC-Bayes generalization bound. To compute it, we evaluate the normalizing constant of equation 1,

$$P(y|x; M) \quad = \quad \int d\omega \, P(y|\omega, x; M)P(\omega; M) \tag{5}$$

$$= \quad \sqrt{\frac{\lambda}{2\pi}} \int d\omega \, e^{-C(\omega; M)}. \tag{6}$$

Notice that the evidence is computed by integrating out the parameters; and consequently it is invariant to the model parameterization. Since this integral is dominated by the region near the minimum $\omega_0$, we can estimate the evidence by Taylor expanding $C(\omega; M) \approx C(\omega_0) + C''(\omega_0)(\omega - \omega_0)^2/2$,

$$P(y|x; M) \quad \approx \quad e^{-C(\omega_0)}\sqrt{\frac{\lambda}{2\pi}} \int d\omega \, e^{-C''(\omega_0)(\omega-\omega_0)^2/2} \tag{7}$$

$$= \quad \exp\left\{-\left(C(\omega_0) + \frac{1}{2}\ln\left(C''(\omega_0)/\lambda\right)\right)\right\}. \tag{8}$$

---

[1]Equivalently, there is an optimal learning rate proportional to the batch size and the training set size.

Within this "Laplace" approximation, the evidence is controlled by the value of the cost function at the minimum, and by the logarithm of the ratio of the curvature about this minimum compared to the regularization constant. Thus far we have considered models of a single parameter; in realistic models with many parameters $P(y|x; M) \approx \lambda^{\frac{p}{2}} e^{-C(\omega_0)} / |\nabla\nabla C(\omega)|^{1/2}_{\omega_0}$, where $|\nabla\nabla C(\omega)|_{\omega_0}$ is the determinant of the Hessian, and $p$ denotes the number of model parameters (Kass & Raftery, 1995). The determinant of the Hessian is simply the product of its eigenvalues, ($\prod_{i=1}^{p} \lambda_i$), and thus,

$$P(y|x; M) \approx \exp\left\{ -\left( C(\omega_0) + \frac{1}{2}\sum_{i=1}^{p} \ln(\lambda_i/\lambda) \right) \right\}. \tag{9}$$

The contribution $(\lambda^{\frac{p}{2}}/|\nabla\nabla C(\omega)|^{1/2}_{\omega_0})$ is often called the "Occam factor", because it enforces Occam's razor; *when two models describe the data equally well, the simpler model is usually better* (Gull, 1988). Minima with low curvature are simple, because the parameters do not have to be fine-tuned to fit the data. Intuitively, the Occam factor describes the fraction of the prior parameter space consistent with the data. Since this fraction is always less than one, we propose to approximate equation 9 away from local minima by only performing the summation over eigenvalues $\lambda_i \geq \lambda$. The evidence can be reframed in the language of information theory, whereby Occam's factor penalizes the amount of information the model must learn about the parameters to accurately model the training data (Hinton & Van Camp, 1993; Achille & Soatto, 2017; Shwartz-Ziv & Tishby, 2017).

In this work, we will compare the evidence against a null model which assumes the labels are entirely random, assigning equal probability to each class. This unusual model has no parameters, and so the evidence is controlled by the likelihood alone, $P(y|x; NULL) = (1/n)^N = e^{-N \ln(n)}$, where $n$ denotes the number of model classes and $N$ the number of training labels. Thus the evidence ratio,

$$\frac{P(y|x; M)}{P(y|x; NULL)} = e^{-E(\omega_0)}, \tag{10}$$

Where $E(\omega_0) = C(\omega_0) + (1/2)\sum_i \ln(\lambda_i/\lambda) - N \ln(n)$ is the log evidence ratio in favor of the null model. Clearly, we should only assign any confidence to the predictions of our model if $E(\omega_0) < 0$.

The evidence supports the intuition that broad minima generalize better than sharp minima, but unlike the curvature it does not depend on the model parameterization. Dinh et al. (2017) showed one can increase the Hessian eigenvalues by rescaling the parameters, but they must simultaneously rescale the regularization coefficients, otherwise the model changes. Since Occam's factor arises from the log ratio, $\ln(\lambda_i/\lambda)$, these two effects cancel out[2]. It is difficult to evaluate the evidence for deep networks, as we cannot compute the Hessian of millions of parameters. Additionally, neural networks exhibit many equivalent minima, since we can permute the hidden units without changing the model. To compute the evidence we must carefully account for this "degeneracy". We argue these issues are not a major limitation, since the intuition we build studying the evidence in simple cases will be sufficient to explain the results of both Zhang et al. (2016) and Keskar et al. (2016).

## 3 BAYES THEOREM AND GENERALIZATION

Zhang et al. (2016) showed that deep neural networks generalize well on training inputs with informative labels, and yet the same model can drastically overfit on the same input images when the labels are randomized; perfectly memorizing the training set. To demonstrate that these observations are not unique to deep networks, let's consider a far simpler model; logistic regression. We form a small balanced training set comprising 800 images from MNIST, of which half have true label "0" and half true label "1". Our test set is also balanced, comprising 5000 MNIST images of zeros and 5000 MNIST images of ones. There are two tasks. In the first task, the labels of both the training and test sets are randomized. In the second task, the labels are informative, matching the true MNIST labels. Since the images contain 784 pixels, our model has just 784 weights and 1 bias.

We show the accuracy of the model predictions on both the training and test sets in figure 1. When trained on the informative labels, the model generalizes well to the test set, so long as it is weakly regularized. However the model also perfectly memorizes the random labels, replicating the observations of Zhang et al. (2016) in deep networks. No significant improvement in model performance

---

[2]Note however that while the evidence itself is invariant to model parameterization, one can find reparameterizations which change the approximate evidence after the Laplace approximation.

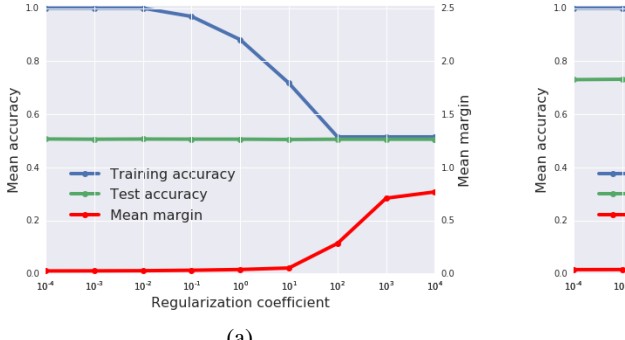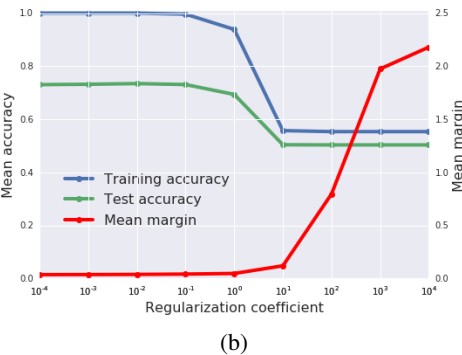

(a)        (b)

Figure 1: Prediction accuracy and mean training set margin as a function of regularization coefficient, for a logistic regression trained on random (a) and informative (b) labels of the same inputs. The weakly regularized model generalizes well on informative labels but memorizes random labels.

is observed as the regularization coefficient increases. For completeness, we also evaluate the mean margin between training examples and the decision boundary. For both random and informative labels, the margin drops significantly as we reduce the regularization coefficient. When weakly regularized, the mean margin is roughly 50% larger for informative labels than for random labels.

Now consider figure 2, where we plot the mean cross-entropy of the model predictions, evaluated on both training and test sets, as well as the Bayesian log evidence ratio defined in the previous section. Looking first at the random label experiment in figure 2a, while the cross-entropy on the training set vanishes when the model is weakly regularized, the cross-entropy on the test set explodes. Not only does the model make random predictions, but it is extremely confident in those predictions. As the regularization coefficient is increased the test set cross-entropy falls, settling at $\ln 2$, the cross-entropy of assigning equal probability to both classes. Now consider the Bayesian evidence, which we evaluate *on the training set*. The log evidence ratio is large and positive when the model is weakly regularized, indicating that the model is exponentially less plausible than assigning equal probabilities to each class. As the regularization parameter is increased, the log evidence ratio falls, but it is always positive, indicating that the model can never be expected to generalize well.

Now consider figure 2b (informative labels). Once again, the training cross-entropy falls to zero when the model is weakly regularized, while the test cross-entropy is high. Even though the model makes accurate predictions, those predictions are overconfident. As the regularization coefficient increases, the test cross-entropy falls below $\ln 2$, indicating that the model is successfully generalizing to the test set. Now consider the Bayesian evidence. The log evidence ratio is large and positive when the model is weakly regularized, but as the regularization coefficient increases, the

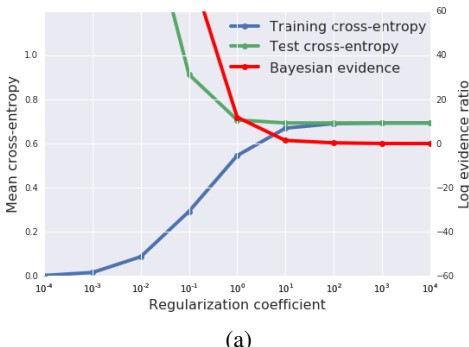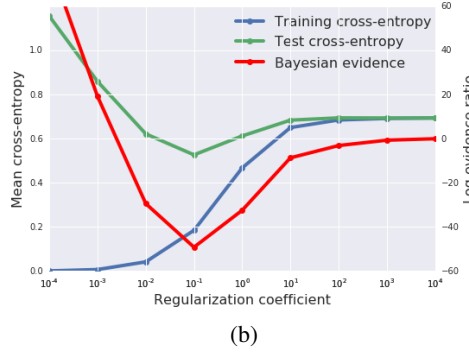

(a)        (b)

Figure 2: The cross-entropy and log evidence ratio, evaluated on random (a) or informative (b) labels. The evidence, evaluated on the training set, is strongly correlated with the test cross-entropy.

log evidence ratio drops *below zero*, indicating that the model is exponentially more plausible than assigning equal probabilities to each class. As we further increase the regularization, the log evidence ratio rises to zero while the test cross-entropy rises to $\ln 2$. Test cross-entropy and Bayesian evidence are strongly correlated, with minima at the same regularization strength.

Bayesian model comparison has explained our results in a logistic regression. Meanwhile, Krueger et al. (2017) showed the largest Hessian eigenvalue also increased when training on random labels in deep networks, implying the evidence is falling. We conclude that Bayesian model comparison is quantitatively consistent with the results of Zhang et al. (2016) in linear models where we can compute the evidence, and qualitatively consistent with their results in deep networks where we cannot. Dziugaite & Roy (2017) recently demonstrated the results of Zhang et al. (2016) can also be understood by minimising a PAC-Bayes generalization bound which penalizes sharp minima.

## 4 BAYES THEOREM AND STOCHASTIC GRADIENT DESCENT

We showed above that generalization is strongly correlated with the Bayesian evidence, a weighted combination of the depth of a minimum (the cost function) and its breadth (the Occam factor). Consequently Bayesians often add isotropic Gaussian noise to the gradient (Welling & Teh, 2011). In appendix A, we show this drives the parameters towards broad minima whose evidence is large. The noise introduced by small batch training is not isotropic, and its covariance matrix is a function of the parameter values, but empirically Keskar et al. (2016) found it has similar effects, driving the SGD away from sharp minima. This paper therefore proposes Bayesian principles also account for the "generalization gap", whereby the test set accuracy often falls as the SGD batch size is increased (holding all other hyper-parameters constant). Since the gradient drives the SGD towards deep minima, while noise drives the SGD towards broad minima, we expect the test set performance to show a peak at an optimal batch size, which balances these competing contributions to the evidence.

We were unable to observe a generalization gap in linear models (since linear models are convex there are no sharp minima to avoid). Instead we consider a shallow neural network with 800 hidden units and RELU hidden activations, trained on MNIST without regularization. We use SGD with a momentum parameter of 0.9. Unless otherwise stated, we use a constant learning rate of 1.0 which does not depend on the batch size or decay during training. Furthermore, we train on just 1000 images, selected at random from the MNIST training set. This enables us to compare small batch to full batch training. We emphasize that we are not trying to achieve optimal performance, but to study a simple model which shows a generalization gap between small and large batch training.

In figure 3, we exhibit the evolution of the test accuracy and test cross-entropy during training. Our small batches are composed of 30 images, randomly sampled from the training set. Looking first at figure 3a, small batch training takes longer to converge, but after a thousand gradient updates a clear generalization gap in model accuracy emerges between small and large training batches. Now consider figure 3b. While the test cross-entropy for small batch training is lower at the end

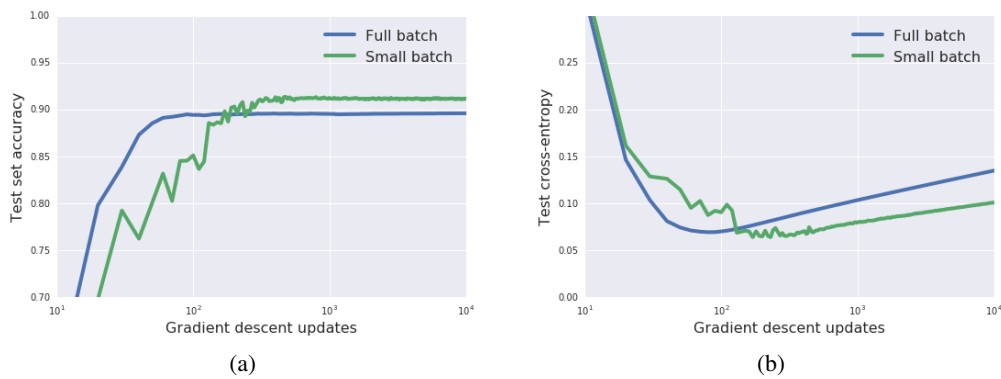

Figure 3: The evolution during training of the test accuracy (a), and the test set cross-entropy (b). Full batches are composed of 1000 images, while small batches comprise 30 images.

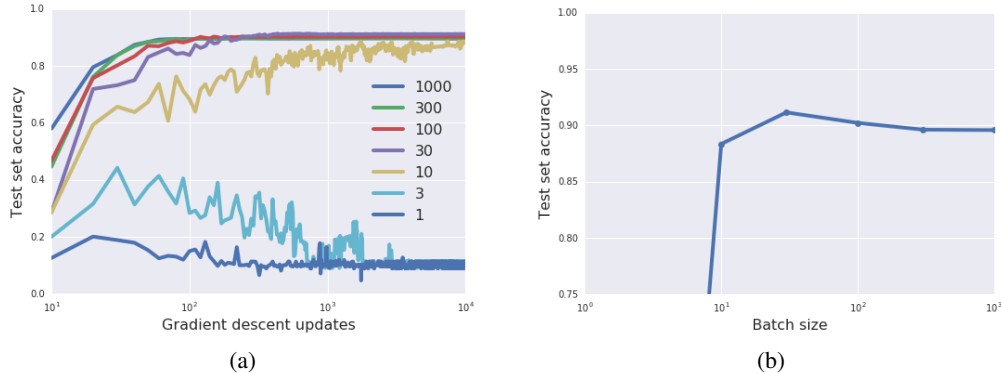

(a)           (b)

Figure 4: The test accuracy for a range of batch sizes, during training (a) and after 10000 steps (b).

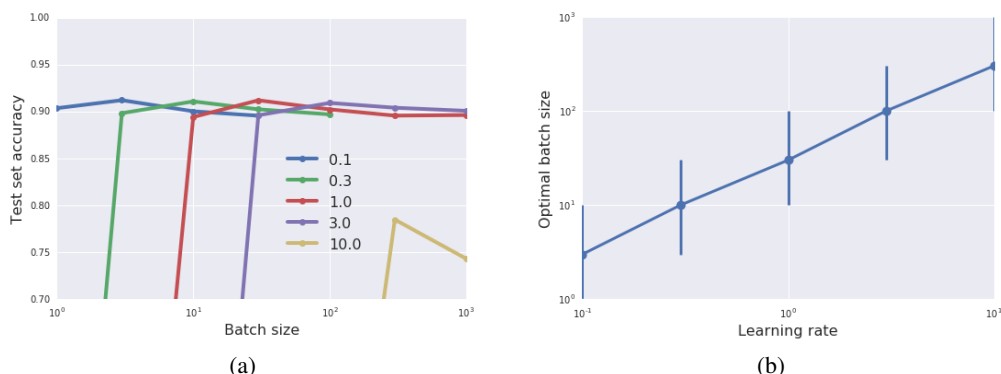

(a)           (b)

Figure 5: a) The test set accuracy as a function of batch size, for a range of learning rates $\epsilon$. The performance peak shifts to the right as we increase $\epsilon$, but the overall performance falls once $\epsilon \gtrsim 3$. b) The best observed batch size is proportional to the learning rate across two orders of magnitude.

of training; the cross-entropy of both small and large training batches is increasing, indicative of over-fitting. Both models exhibit a minimum test cross-entropy, although after different numbers of gradient updates. Intriguingly, we show in appendix B that the generalization gap between small and large batch training shrinks significantly when we introduce L2 regularization.

From now on we focus on the test set accuracy (since this converges as the number of gradient updates increases). In figure 4a, we exhibit training curves for a range of batch sizes between 1 and 1000. We find that the model cannot train when the batch size $B \lesssim 10$. In figure 4b we plot the mean test set accuracy after 10000 training steps. A clear peak emerges, indicating that there is indeed an optimum batch size which maximizes the test accuracy, consistent with Bayesian intuition. The results of Keskar et al. (2016) focused on the decay in test accuracy above this optimum batch size.

## 5   STOCHASTIC DIFFERENTIAL EQUATIONS AND THE SCALING RULES

We showed above that the test accuracy peaks at an optimal batch size, *if one holds the other SGD hyper-parameters constant*. We argued that this peak arises from the tradeoff between depth and breadth in the Bayesian evidence. However it is not the batch size itself which controls this tradeoff, but the underlying scale of random fluctuations in the SGD dynamics. We now identify this SGD "noise scale", and use it to derive three scaling rules which predict how the optimal batch size depends on the learning rate, training set size and momentum coefficient. A gradient update,

$$\Delta\omega \;=\; -\frac{\epsilon}{N}\left(\frac{dC}{d\omega} + \left(\frac{d\hat{C}}{d\omega} - \frac{dC}{d\omega}\right)\right), \tag{11}$$

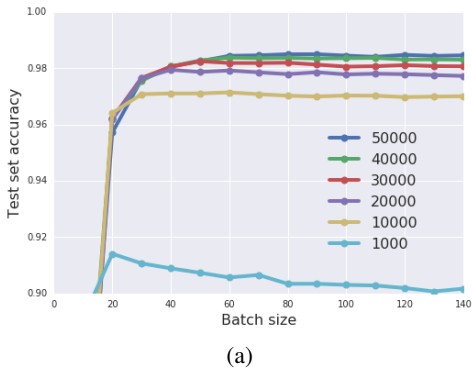
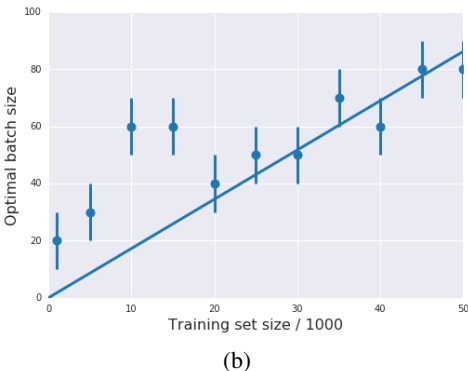

(a)                                                        (b)

Figure 6: a) The test accuracy as a function of batch size, for a range of training set sizes. To reduce noise, we average each curve over five experiments. The performance peak shift to the right as we increase the size of the training set. Unsurprisingly, the overall model performance also improves. b) The best observed batch size is proportional to the size of the training set once $N \gtrsim 20000$.

where $\epsilon$ is the learning rate, $N$ the training set size, $\frac{dC}{d\omega} = \sum_{i=1}^{N} \frac{dC_i}{d\omega}$ the true gradient, and $\frac{d\hat{C}}{d\omega} = \frac{N}{B} \sum_{i=1}^{B} \frac{dC_i}{d\omega}$ the estimated gradient evaluated on a mini-batch. The expected gradient of a single example, $\left\langle \frac{dC_i}{d\omega} \right\rangle = \frac{1}{N} \frac{dC}{d\omega}$, while $\left\langle \frac{dC_i}{d\omega} \frac{dC_j}{d\omega} \right\rangle = \left( \frac{1}{N} \frac{dC}{d\omega} \right)^2 + F(\omega)\delta_{ij}$. $F(\omega)$ is a matrix describing the gradient covariances, which are a function of the current parameter values. We adopt the central limit theorem and model the gradient error $\alpha = \left( \frac{d\hat{C}}{d\omega} - \frac{dC}{d\omega} \right)$ with Gaussian random noise (We discuss this approximation briefly in appendix C). It is easy to show that $\langle \alpha \rangle = 0$, while $\langle \alpha^2 \rangle = N(\frac{N}{B} - 1)F(\omega)$. Typically $N \gg B$, such that $\langle \alpha^2 \rangle \approx N^2 F(\omega)/B$. To continue, we interpret equation 11 as the discrete update of a stochastic differential equation (Li et al., 2017; Gardiner, 1985),

$$\frac{d\omega}{dt} = -\frac{dC}{d\omega} + \eta(t), \tag{12}$$

Where $t$ is a continuous variable, $\eta(t)$ represents noise, $\langle \eta(t) \rangle = 0$ and $\langle \eta(t)\eta(t') \rangle = gF(\omega)\delta(t-t')$. The constant $g$ controls the scale of random fluctuations in the dynamics. To relate this differential equation to the SGD, we compute a gradient update $\Delta\omega = \int_0^{\epsilon/N} \frac{d\omega}{dt} dt = -\frac{\epsilon}{N} \frac{dC}{d\omega} + \int_0^{\epsilon/N} \eta(t)dt$. Finally, to measure g, we equate the variance in this gradient update to the variance in equation 11,

$$\left( \frac{\epsilon}{N} \right)^2 \langle \alpha^2 \rangle = \epsilon^2 (\frac{N}{B} - 1)F(\omega)/N$$

$$= \left\langle \left( \int_0^{\epsilon/N} dt\, \eta(t) \right)^2 \right\rangle = \int_0^{\epsilon/N} dt \int_0^{\epsilon/N} dt' \, \langle \eta(t)\eta(t') \rangle = \epsilon gF(\omega)/N. \tag{13}$$

Rearranging, the SGD noise scale $g = \epsilon(\frac{N}{B} - 1) \approx \epsilon N/B$. The noise scale falls when the batch size increases, consistent with our earlier observation of an optimal batch size $B_{opt}$ while holding the other hyper-parameters fixed. Notice that one would equivalently observe an optimal learning rate if one held the batch size constant. A similar analysis of the SGD was recently performed by Mandt et al. (2017), although their treatment only holds near local minima where the covariances $F(\omega)$ are stationary. Our analysis holds throughout training, which is necessary since Keskar et al. (2016) found that the beneficial influence of noise was most pronounced at the start of training.

When we vary the learning rate or the training set size, we should keep the noise scale fixed, which implies that $B_{opt} \propto \epsilon N$. In figure 5a, we plot the test accuracy as a function of batch size after $(10000/\epsilon)$ training steps, for a range of learning rates. Exactly as predicted, the peak moves to the right as $\epsilon$ increases. Additionally, the peak test accuracy achieved at a given learning rate does not begin to fall until $\epsilon \sim 3$, indicating that there is no significant discretization error in integrating the stochastic differential equation below this point. Above this point, the discretization error begins to dominate and the peak test accuracy falls rapidly. In figure 5b, we plot the best observed batch size as a function of learning rate, observing a clear linear trend, $B_{opt} \propto \epsilon$. The error bars indicate the distance from the best observed batch size to the next batch size sampled in our experiments.

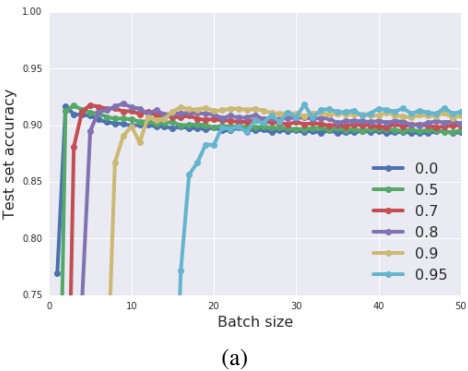
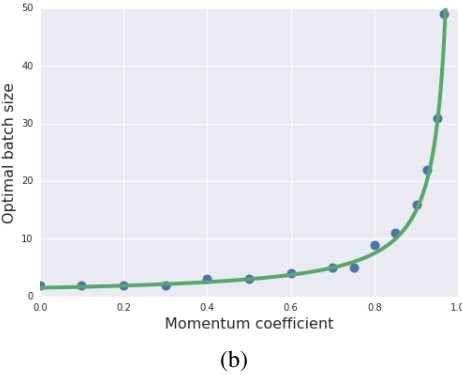

(a)                          (b)

Figure 7: a) The test set accuracy as a function of batch size for a range of momentum coefficients. As expected, the peak moves to the right as the momentum coefficient increases. b) The best observed batch size for a range of momentum coefficients. The green curve exhibits the scaling rule.

This scaling rule allows us to increase the learning rate with no loss in test accuracy and no increase in computational cost, simply by simultaneously increasing the batch size. We can then exploit increased parallelism across multiple GPUs, reducing model training times (Goyal et al., 2017). A similar scaling rule was independently proposed by Jastrzebski et al. (2017) and Chaudhari & Soatto (2017), although neither work identifies the existence of an optimal noise scale. A number of authors have proposed adjusting the batch size adaptively during training (Friedlander & Schmidt, 2012; Byrd et al., 2012; De et al., 2017), while Balles et al. (2016) proposed linearly coupling the learning rate and batch size within this framework. In Smith et al. (2017), we show empirically that decaying the learning rate during training and increasing the batch size during training are equivalent.

In figure 6a we exhibit the test set accuracy as a function of batch size, for a range of training set sizes after 10000 steps ($\epsilon = 1$ everywhere). Once again, the peak shifts right as the training set size rises, although the generalization gap becomes less pronounced as the training set size increases. In figure 6b, we plot the best observed batch size as a function of training set size; observing another linear trend, $B_{opt} \propto N$. This scaling rule could be applied to production models, progressively growing the batch size as new training data is collected. We expect production datasets to grow considerably over time, and consequently large batch training is likely to become increasingly common.

Finally, in appendix D we extend our analysis to SGD with momentum, identifying the noise scale, $g \approx \frac{\epsilon N}{B(1-m)}$, where $m$ denotes the momentum coefficient. Notice that this reduces to the noise scale of conventional SGD as $m \to 0$. When $m > 0$, we obtain an additional scaling rule $B_{opt} \propto 1/(1-m)$. This scaling rule predicts that the optimal batch size will increase when the momentum coefficient is increased. In figure 7a we plot the test set performance as a function of batch size after 10000 gradient updates ($\epsilon = 1$ everywhere), for a range of momentum coefficients. In figure 7b, we plot the best observed batch size as a function of the momentum coefficient, and fit our results to the scaling rule above; obtaining remarkably good agreement. We propose a simple heuristic for tuning the batch size, learning rate and momentum coefficient in appendix E.

## 6 CONCLUSIONS

Just like deep neural networks, linear models which generalize well on informative labels can memorize random labels of the same inputs. These observations are explained by the Bayesian evidence, which is composed of the cost function and an "Occam factor". The Occam factor penalizes sharp minima but it is invariant to changes in model parameterization. Mini-batch noise drives SGD away from sharp minima, and therefore there is an optimum batch size which maximizes the test accuracy. Interpreting SGD as the discretization of a stochastic differential equation, we predict this optimum batch size should scale linearly with both the learning rate and the training set size, $B_{opt} \propto \epsilon N$. We derive an additional scaling rule, $B_{opt} \propto 1/(1-m)$, between the optimal batch size and the momentum coefficient. We verify these scaling rules empirically and discuss their implications.

ACKNOWLEDGMENTS

We thank Pieter-Jan Kindermans, Prajit Ramachandran, Jascha Sohl-Dickstein, Jon Shlens, Kevin Murphy, Samy Bengio, Yasaman Bahri and Saeed Saremi for helpful comments on the manuscript.

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

## A  BAYESIAN POSTERIOR SAMPLING AND LANGEVIN DYNAMICS

Instead of minimizing the cost function, Bayesian usually prefer to sample parameter values from the posterior (MacKay, 1992),

$$P(\omega|y, x; M) \propto e^{-C(\omega; M)}, \tag{14}$$

where $C(\omega; M)$ is the regularized summed cost function, as shown in section 2 of the main text. It is well known that one can sample this posterior by simulating the overdamped Langevin equation (Gardiner, 1985), which is described by the stochastic differential equation,

$$\frac{d\omega}{dt} = -\frac{dC}{d\omega} + \eta(t), \tag{15}$$

where $t$ is a continuous variable, and $\eta(t)$ describes Gaussian noise with mean $\langle \eta(t) \rangle = 0$ and variance $\langle \eta(t)\eta(t') \rangle = 2TI\delta(t - t')$. The matrix $I$ denotes the identity, while $T$ is the "temperature". Notice this Langevin equation is extremely similar to the stochastic differential equation of SGD, discussed in section 5 of the main text. Indeed, if the gradient covariances $F(\omega)$ were stationary and proportional to the identity, then the SGD would integrate an overdamped Langevin equation with temperature proportional to the SGD noise scale $g$. As $t \to \infty$, the probability of sampling any particular parameter vector $\omega$ from the Langevin equation, $P(\omega, t \to \infty) \propto e^{-C/T}$.

We obtain posterior samples if $T = 1$. In order to draw posterior samples in practice, we repeatedly integrate the Langevin equation (at temperature $T = 1$), over a finite step $t \to t + \epsilon/N$,

$$\Delta\omega = -\frac{\epsilon}{N}\frac{dC}{d\omega} + \int_t^{t + \frac{\epsilon}{N}} \eta(t)dt \tag{16}$$

$$= -\frac{\epsilon}{N}\frac{dC}{d\omega} + \alpha, \tag{17}$$

where $\alpha$ denotes a Gaussian random variable with mean $\langle\alpha\rangle = 0$ and variance $\langle\alpha^2\rangle = 2\epsilon I/N$, which introduces isotropic noise to the gradient update as described in section 4 of the main text. Note that, since $C(\omega; M)$ denotes the summed cost function, we chose to scale our step size by the training set size $N$. This also matches our treatment of SGD in section 5 of the main text. The larger the step size $\epsilon$, the greater the discretization error, but if $\epsilon$ is sufficiently small and we iterate equation 17 sufficiently many times, we will obtain valid samples from the posterior.

Since the probability of sampling any given parameter vector $\omega$ is proportional to the posterior, the probability of sampling a parameter vector belonging to any given local minimum is proportional to the integral of the posterior over the bowl of attraction $D$ which surrounds that minimum.

$$P(\omega \in D, t \to \infty) \propto \int_D d\omega \; e^{-C(\omega;M)}. \tag{18}$$

Meanwhile we showed in section 2 of the main text that the evidence in favor of a model is proportional to the integral of the posterior over all parameter space.

$$P(y|x; M) \propto \int d\omega \; e^{-C(\omega;M)}. \tag{19}$$

As we discussed, this evidence is dominated by the contributions to the integral near local minima. In a convex model, there is only one such minimum; which allows us to accurately estimate the model evidence. Meanwhile, in non-convex models, there are many such minima, and so we can instead define the evidence as a sum over local evidences in favor of each minimum,

$$P(y|x; M) = \sum_i P(y|x, \omega \in D_i; M), \tag{20}$$

where we define the evidence in favor of a minimum as the integral over the local bowl of attraction,

$$P(y|x, \omega \in D; M) \propto \int_D d\omega \; e^{-C(\omega;M)}. \tag{21}$$

Since the combined bowls of attraction of all the minima perfectly tile the entire parameter space, equations 19 and 20 are equivalent. Meanwhile, equating equations 18 and 21 we find that, when one performs Bayesian posterior sampling, the probability of sampling the parameter vector from a local minimum is proportional to the evidence in favor of that minimum. This demonstrates that bayesian posterior samples are biased in favor of local minima whose evidence is large, which explains why a single posterior sample $\omega_p$ often achieves lower test error than the cost function minimum $\omega_0$.

## B   THE EFFECT OF REGULARIZATION ON THE GENERALIZATION GAP

In the experiments of section 4 of the main text, the L2 regularization coefficient $\lambda = 0$. In figure 8, we plot the evolution of the training curves when $\lambda = 0.1$, for both small batch and full batch

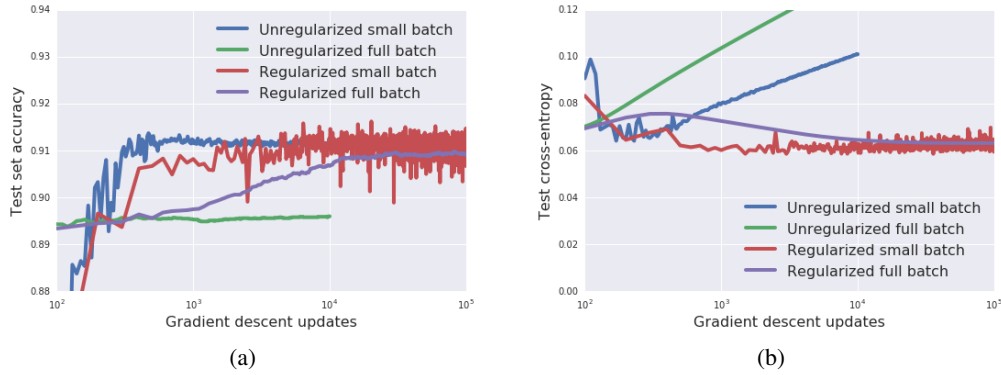

(a)                                           (b)

Figure 8: The mean test accuracy (a) and the mean test cross-entropy (b) of a regularized model during training. While full batch training takes longer to converge, it achieves similar performance at long times. The noise inherent in small batch training causes the performance to fluctuate.

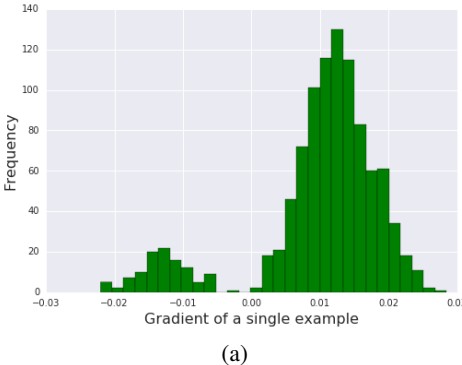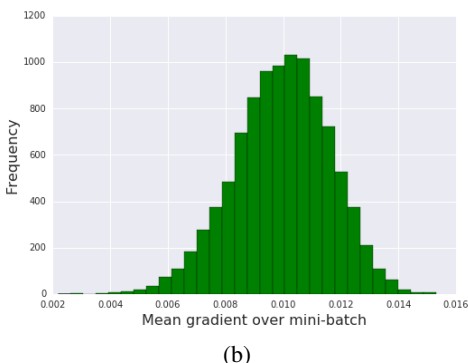

(a)                                                        (b)

Figure 9: The gradient distribution of a randomly selected parameter in the softmax layer, when measured over a single training example (a), and when averaged over mini-batches of 30 images (b).

training. Excluding the regularization parameter, these experiments are identical to figure 3. To our surprise, regularized full batch training took longer to converge than small batch training. In another surprise, regularization significantly reduced the size of the generalization gap. While large batch regularized training achieves slightly lower test set accuracy than unregularized small batch training, it also achieves lower test cross-entropy. The test cross-entropy of our regularized models does not degrade after many gradient updates, removing the need for early stopping.

## C   THE GAUSSIAN APPROXIMATION TO THE MINI-BATCH ERROR

In section 5 of the main text, we approximated the difference between the full batch gradient and the mini-batch gradient estimate, $\alpha = (\frac{d\hat{C}}{d\omega} - \frac{dC}{d\omega})$, by a Gaussian random variable. This enabled us to derive the scaling rules, which we verified empirically. We motivated this assumption by reference to the central limit theorem, which states that the gradient error will tend towards Gaussian noise as $\{N \to \infty, B \to \infty, B \ll N\}$, so long as the distribution of gradients over individual training examples does not have heavy tails. In practice neither N nor B is infinite, and the gradient distribution may be heavy tailed, especially when gradients are sparse. Nonetheless the central limit theorem tends to be surprisingly robust in practice, and is consequently widely used.

It is beyond the scope of this work to perform a thorough study of the gradient noise distribution in deep networks. However as a brief proof of principle, we present the distribution of the gradient immediately after random initialization in figure 9, for the shallow neural network discussed in sections 4 and 5 of the main text. In figure 9a, we present the distribution over the individual training examples, of the gradient of a single matrix element in the softmax output layer, chosen randomly. The distribution is double peaked and clearly not Gaussian. However in figure 7b, we plot the distribution of the gradient of the same matrix element, when averaged over randomly sampled mini-batches of 30 images (without replacement). A single peak emerges, and while the distribution is still slightly skewed, it is clearly already approaching the Gaussian limit. We conclude that the Gaussian approximation is likely to be reasonable for commonly used mini-batch sizes.

## D   DERIVING THE SCALING RULES FOR SGD WITH MOMENTUM

Momentum simulates a generalized Langevin equation (with structured fluctuations),

$$\frac{d^2\omega}{dt^2} = -\lambda\frac{d\omega}{dt} - \frac{dC}{d\omega} + \eta(t). \tag{22}$$

$\lambda$ is the "damping coefficient" and $\eta(t)$ describes Gaussian noise, whose statistics $\langle\eta(t)\rangle = 0$ and $\langle\eta(t)\eta(t')\rangle = g\lambda F(w)\delta(t-t')$. As before, the coefficient $g$ describes the scale of random fluctuations in the dynamics, and $F(\omega)$ describes the gradient covariances between parameters. We include a factor of $\lambda$ in the noise variance to satisfy the fluctuation-dissipation theorem, which states that

we can vary the damping coefficient without changing the probability of sampling any particular configuration of parameters in the limit $t \to \infty$, if we proportionally increase the noise variance.

To relate this Langevin equation to the usual momentum equations, we first re-express it as two coupled first order differential equations,

$$\frac{dp}{dt} = -\lambda p - \frac{dC}{d\omega} + \eta(t), \tag{23}$$

$$\frac{d\omega}{dt} = p. \tag{24}$$

Integrating over a single step $\Delta t/N$,

$$\Delta p = -(\lambda \Delta t/N)p - \frac{\Delta t}{N}\frac{dC}{d\omega} + \eta, \tag{25}$$

$$\Delta \omega = p\Delta t/N. \tag{26}$$

Where now $\langle \eta \rangle = 0$ and $\langle \eta^2 \rangle = g\Delta t\lambda F(w)/N$. We define the accumulation $A = p/\Delta t$,

$$\Delta A = -(\lambda \Delta t/N)A - \frac{1}{N}\frac{dC}{d\omega} + \frac{\eta}{\Delta t}, \tag{27}$$

$$\Delta \omega = (\Delta t)^2 A/N. \tag{28}$$

These equations can be compared to the TensorFlow update equations for momentum,

$$\Delta A = (m-1)A - \frac{1}{N}\left(\frac{dC}{d\omega} + \alpha\right), \tag{29}$$

$$\Delta \omega = \epsilon A. \tag{30}$$

Where $\alpha = \left(\frac{d\hat{C}}{d\omega} - \frac{dC}{d\omega}\right)$ denotes the error in the gradient update. As discussed in the main text, we can approximate this error as Gaussian noise with statistics $\langle \alpha \rangle = 0$ and $\langle \alpha^2 \rangle \approx N^2 F(\omega)/B$. Equations 27 and 28 match equations 29 and 30 if the step size $\epsilon = (\Delta t)^2/N$, and the momentum parameter $m = (1 - \lambda \Delta t/N)$. Finally we equate the noise by setting $\langle \alpha^2 \rangle/N^2 = \langle \eta^2 \rangle/(\Delta t)^2$, and solve for the noise scale $g$ to obtain,

$$g \approx \frac{\Delta t N}{B\lambda} = \frac{(\Delta t)^2}{B(1-m)} \tag{31}$$

$$= \frac{\epsilon N}{B(1-m)}. \tag{32}$$

As observed in the main text, if we wish to keep the scale of random fluctuations constant, then we should scale the batch size $B \propto \epsilon N$. We also predict an additional scaling relation between the batch size and the momentum parameter, $B \propto 1/(1-m)$. Note that one can also interpret $\epsilon_{eff} = \epsilon/(1-m)$ as the "effective learning rate".

## E   HOW TO ACHIEVE LARGE BATCH TRAINING

Here we propose a simple heuristic for tuning the batch size, learning rate and momentum parameter; in order to maximize both test accuracy and batch size (enabling parallel training across many machines). Note that this is only worthwhile if one expects to retrain a model many times.

1. Set the learning rate to 0.1 and the momentum coefficient to 0.9. Run experiments at a range of batch sizes on a logarithmic scale, and identify the optimal batch size which maximizes the validation set accuracy. If training is not stable, reduce the learning rate, and repeat.

2. Repeatedly increase the batch size by a factor of 3, while scaling the learning rate $\epsilon \propto B$, until the validation set accuracy starts to fall. Then repeatedly increase the batch size by a factor of 3, while scaling the momentum coefficient $(1 - m) \propto 1/B$, until either the validation set accuracy falls or the batch size reaches the limits of your hardware.

3. Having identified the final learning rate and momentum parameter, retune the batch size on a linear scale in the local neighborhood of the current batch size.

We believe that this simple procedure will increase the test accuracy, reduce the cost of tuning hyper-parameters, and significantly reduce the final number of gradient updates required to train a model.

