# OpenReview forum: "A Bayesian Perspective on Generalization and Stochastic Gradient Descent"
_ICLR.cc/2018/Conference — Accept (Poster)_

### Official Review · AnonReviewer3 · 2017-11-27
**Bottom line: The paper may contribute to the current discussion of the Zhang et al 2016 paper, but I feel  it does not make a significant contribution to the state of knowledge in machine learning. On top of that, I feel the execution of the paper leaves much to be desired.**

**Rating:** 3
**Confidence:** 4

**Review:**

The paper takes a recent paper of Zhang et al 2016 as the starting point to investigate the generalization capabilities of models trained by stochastic gradient descent. The main contribution are scaling rules that relate the batch size k used in SGD with the learning rate \epsilon, most notably \epsilon/k = const for optimal scaling.

First of all, I have to say that the paper is very much focussed on the aforementioned paper, its experiments as well as its (partially speculative) claims. This, in my opinion, is a biased and limited starting point, which ignores much of the literature in learning theory.

Chapter 2 provides a sort of a mini-tutorial to (Bayesian) model selection based on standard Bayes factors. I find this of limited usefulness. First of all, I find the execution poor in the details:
(i) Why is \omega limited to a scalar? Nothing major really depends on that. Later the presentation switches to a more general case.
(ii) What is a one-hot label? "One-hot" is the encoding of a categorical label.
(iii) In which way is a Gaussian prior uncorrelated, if there is just a scalar random variable?
(iv) How can one maximize a probability density function?
(v) Why is an incorrect "pseudo"-set notation used instead of the correct vectorial one?
(vi) "Exponentially large", "reasonably prior" model etc. is very vague terminology
(vii) No real credit is given for the Laplace approximation presented up to Eq. 10. For instance, why not refer to the seminal paper by Kass & Raferty? Why spend so much time on a step-by-step derivation anyway, as this is all "classic" and has been carried out many times before (in a cleaner write-up)?
(viii) "P denotes the number of model parameters" (I guess it should be a small p? hard to decipher)
(ix) Usually, one should think of the Laplace approximation and the resulting Bayes factors more in terms of a "volume" of parameters  close to the MAP estimate, which is what the matrix determinant expresses, more than any specific direction of "curvature".

Chapter 3 constructs a simple example with synthetic data to demonstrate the effect of Bayes factors. I feel the discussion to be too much obsessed by the claims made in Zhang et al 2016 and in no way suprising. In fact, the "toy" example is so much of a "toy" that I am not sure what to make of it. Statistics has for decades successfully used criteria for model selection, so what is this example supposed to proof (to whom?).

Chapter 4 takes the work of Mandt et al as a starting point to understand how SGD with constant step size effectively can be thought of as gradient descent with noise, the amplitude of which is controlled by the step size and the mini-batch size. Here, the main goal is to use evidence-based arguments to distinguish good from poor local minima. There is some experimental evidence presented on how to resolve the tradeoff between too much noise (underfitting) and too little (overfitting).

Chapter 5 takes a stochastic differential equation as a starting point. I see several issues:
(i) It seems that you are not doing much with a SDE, as you diredctly jump to the discretized version (and ignore discussions of it's discretization). So maybe one should not feature the term SDE so prominently.
(ii) While it is commonly done, it would be nice to get some insights on why a Gaussian approx. is a good assumption. Maybe you can verify this experimentally (as much of the paper consists of experimental findings)
(iii) Eq. 13. Maybe you want this form to indicate a direction you want to move towards,  by I find adding and subtracting the gradient in itself not a very interesting manner of illustartion.
(iv) I am not sure in whoch way g is "measured", but I guess you are determining it by comparing coefficients.
(v) I am confused by the B_opt \propto \eps statement. It seems you are scaling to mini-batrch gradient to be in expectation equal to the full gradient (not normalized by N), e.g. it scales ~N. Now, if we think of a mini-batch as being a batched version of single pattern updates, then clearly the effective step length should scale with the batch size, which - because of the batch size normalization with N/B - means \epsilon needs to scale with B. Maybe there is something deeper going on here, but it is not obvious to me.
(vi) The argument why B ~ N is not clear to me. Is there one or are just making a conjecture?

Bottom line: The paper may contribute to the current discussion of the Zhang et al 2016 paper, but I feel  it does not make a significant contribution to the state of knowledge in machine learning. On top of that, I feel the execution of the paper leaves much to be desired.

---

> ### Author Response · Authors · 2017-12-07
> **Response to review**
>
> We are very grateful for the helpful comments provided, however we feel that the score attached to this review does not recognize the many significant contributions of our work. The bulk of the specific comments raised are relatively minor and easily addressed. We would like to encourage the reviewer to reconsider.
>
> Our work demonstrates that a number of active debates in deep learning can be resolved by well-established Bayesian principles.
>
> 1) Zhang et al. showed deep networks which generalize well on real training data can still memorize random labellings of the same inputs. In section 3 we observe exactly the same phenomenon in an over-parameterized linear model, and demonstrate that this phenomenon is easily and quantitatively explained by Bayesian model comparison. Taken together, these results demonstrate that deep learning does not require rethinking generalization.
>
> We recognize that Zhang et al.'s claims regarding learning theory are disputed and we will make this clear when we update the manuscript. However the paper received the best paper award and inspired many follow-up works. Since the reviewer was unhappy with our synthetic data experiments, we will replace these by a linear model trained to distinguish real '0's and '1's from the MNIST dataset. We have already run these experiments and the results are identical.
>
> 2) No previous author has explicitly demonstrated that there is an optimum batch size (at constant learning rate) and this clarifies earlier work by Keskar et al. (ICLR 2017). We argue that these results are easily understood as arising from the competition between the depth and breadth of minima in the Bayesian evidence.
>
> 3) It is not the batch size itself which controls this competition; it is the magnitude of random fluctuations in the SGD dynamics. We derive a novel closed form expression for this “noise scale” which holds even when the gradient covariances between parameters are anisotropic and non-stationary. This closed form expression predicts three scaling rules, which we verify empirically. These scaling rules can be used to increase the optimal batch size without reducing the test accuracy; enabling parallelized training with large mini-batches.
>
> In response to the remaining specific comments:
> Section 2)
> We state in the opening paragraph that our derivation “closely [follows] the seminal work of David MacKay (1992)”. We are happy to cite Kass and Raferty (1995). We know this Laplacian treatment is “classic”. We include it because it is central to all of our observations which follow, and because most researchers in deep learning have never seen it.
>
> i) Mackay first considers a single scalar parameter, we are following his approach. It is easy to replace the 2nd derivative by the Hessian at the end.
> ii) The cross entropy measures a distance between label distributions. By one-hot label we mean that the example has a single unique label.
> iii) This is a typo and we will remove it.
> iv) The minimum of the cost function corresponds to the maximum of the posterior. We will make this clearer.
> v) We are happy to change this as requested.
> vi) We will edit the text to clarify. The evidence ratio will grow exponentially in the number of training examples, while the contribution from the prior is constant.
> vii) We gave credit to Mackay, who we believe was the first to apply Bayesian model comparison to neural networks.
> viii) We are happy to change this to small p.
> ix) We interpret the evidence in terms of curvature to show Bayesian principles resolve the current debate in deep learning regarding sharp minima. Crucially, the Bayesian evidence is invariant to model parameterization, resolving the objections of Dinh et al. (ICML 2017). We believe this clarification is important.
>
> Section 5)
> i) The effect of discretization is clear from figure 5a, for which there is no significant discretization error until the learning rate ~ 3 (ie the peak test accuracy is constant as we increase the learning rate). After this point the test accuracy falls rapidly. We will discuss this discretization error more explicitly when we update the manuscript. We used the SDE to derive the three scaling rules.
> ii) We will add a discussion of the Gaussian approximation to the appendix. However we note that we already verified all of our key practical predictions empirically.
> iii) We feel this makes the following SDE derivation clearer.
> iv) In equation 15, we equate the variance of a gradient update, to the variance of the SDE, integrated over the duration of the update. Rearranging, one obtains g = \eps N / B.
> v) In equation 13, we normalize the gradient update by the number of training examples N. The scaling rules apply to the mean gradient, not the summed gradient.
> vi) Bayesian arguments explicitly support the B ~ N scaling rule. For instance, in Langevin posterior sampling, the magnitude of the noise added to the gradient update is inversely proportional to N. We will make this connection clearer.

---

### Official Review · AnonReviewer1 · 2017-11-27
**Insightful paper, proposing use of "Bayesian evidence" as a way of evaluating the simplicity and generalizability of a neural network model.**

**Rating:** 7
**Confidence:** 4

**Review:**

Summary:
This paper presents a very interesting perspective on why deep neural networks may generalize well, in spite of their high capacity (Zhang et al, 2017). It does so from the perspective of "Bayesian model comparison", where two models are compared based on their "marginal likelihood" (aka, their "evidence" --- the expected probability of the training data under the model, when parameters are drawn from the prior).  It first shows that a simple weakly regularized (linear) logistic regression model over 200 dimensional data can perfectly memorize a random training set with 200 points, while also generalizing well when the class labels are not random (eg, when a simple linear model explains the class labels); this provides a much simpler example of a model generalizing well in spite of high capacity, relative to the experiments presented by Zhang et al (2017). It shows that in this very simple setting, the "evidence" of a model correlates well with the test accuracy, and thus could explain this phenomena (evidence is low for model trained on random data, but high for model trained on real data).

The paper goes on to show that if the evidence is approximated using a second order Taylor expansion of the cost function around a minimia $w_0$, then the evidence is controlled by the cost at the minimum, and by the logarithm of the ratio of the curvature at the minimum compared to the regularization constant (eg, standard deviation of gaussian prior).  Thus, Bayesian evidence prefers minima that are both deep and broad.  This provides a way of comparing models in a way which is independent of the model parametrization (unfortunately, however, computing the evidence is intractable for large networks). The paper then discusses how SGD can be seen as an algorithmic way of finding minima with large "evidence" --- the "noise" in the gradient estimation helps the model avoid "sharp" minima, while the gradient helps the model find "deep" minima.  The paper shows that SGD can be understood using stochastic differential equations, where the noise scale is approximately aN/((1-m)B) (a = learning rate, N = size of training set, B = batch size, m = momentum).  It argues that because there should be an optimal noise scale (which maximizes test performance), the batch size should be taken proportional to the learning rate, as well as the training set size, and proportional to 1/(1-m).  These scaling rules are confirmed experimentally (DNN trained on MNIST).  Thus, this Bayesian perspective can also help explain the observation that models trained with smaller batch sizes (noisier gradient estimates) often generalize better than those with larger batch sizes (Kesker et al, 2016). These scaling rules provide guidance on how to increase the batch size, which is desirable for increasing the parralelism of SGD training.

Review:
Quality: The quality of the work is high.  Experiments and analysis are both presented clearly.

Clarity: The paper is relatively clear, though some of the connections between the different parts of the paper felt unclear to me:
1) It would be nice if the paper were to explain, from a theoretical perspective, why large evidence should correspond to better generalization, or provide an overview of the work which has shown this (eg, Rissanen, 1983).
2) Could margin-based generalization bounds explain the superior generalization performance of the linear model trained on random vs. non-random data?  It seems to me that the model trained on meaningful data should have a larger margin.
3) The connection between the work on Bayesian evidence, and the work on SGD, felt very informal. The link seems to be purely intuitive (SGD should converge to minima with high evidence, because its updates are noisy).  Can this be formalized?  There is a footnote on page 7 regarding Bayesian posterior sampling -- I think this should be brought into the body of the paper, and explained in more detail.
4) The paper does not give any background on stochastic differential equations, and why there should be an optimal noise scale 'g', which remains constant during the stochastic process, for converging to a minima with high evidene.  Are there any theoretical results which can be leveraged from the stochastic processes literature? For example, are there results which prove anything regarding the convergence of a stochastic process under different amounts of noise?
5) It was unclear to me why momentum was used in the MNIST experiments.  This seems to complicate the experimental setting.  Does the generalization gap not appear when no momentum is used?  Also, why is the same learning rate used for both small and large batch training for Figures 3 and 4?  If the learning rate were optimized together with batch size (eg, keeping aN/B constant), would the generalization gap still appear?  Figure 5a seems to suggest that it would not appear (peaks appear to all have the same test accuracy).
6) It was unclear to me whether the analysis of SGD as a stochastic differential equation with noise scale aN/((1-m)B) was a contribution of this paper.  It would be good if it were made clearer which part of the mathematical analysis in sections 2 and 5 are original.
7) Some small feedback: The notation $< x_i > = 0$ and $< x_i^2 > = 1$ is not explained.  Is each feature being normalized to be zero mean, unit variance, or is each training example being normalized?

Originality: The works seems to be relatively original combination of ideas from Bayesian evidence, to deep neural network research.  However, I am not familiar enough with the literature on Bayesian evidence, or the literature on sharp/broad minima, and their generalization properties, to be able to confidently say how original this work is.

Significance: I believe that this work is quite significant in two different ways:
1) "Bayesian evidence" provides a nice way of understanding why neural nets might generalize well, which could lead to further theoretical contributions.
2) The scaling rules described in section 5 could help practitioners use much larger batch sizes during training, by simultaneously increasing the learning rate, the training set size, and/or the momentum parameter.  This could help parallelize neural network training considerably.

Some things which could limit the significance of the work:
1) The paper does not provide a way of measuring the (approximate) evidence of a model.  It simply says it is prohibitively expensive to compute for large models.  Can the "Gaussian approximation" to the evidence (equation 10) be approximated efficiently for large neural networks?
2) The paper does not prove that SGD converges to models of high evidence, or formally relate the noise scale 'g' to the quality of the converged model, or relate the evidence of the model to its generalization performance.

Overall, I feel the strengths of the paper outweight its weaknesses.  I think that the paper would be made stronger and clearer if the questions I raised above are addressed prior to publication.

---

> ### Author Response · Authors · 2017-12-13
> **Response to review**
>
> We thank the referee for their positive assessment of our work.
> Regarding the originality of our work, we believe our paper makes three main contributions:
>
> 1) We show that well-established Bayesian principles can resolve a number of active debates in the deep learning community (generalization/sharp minima/reparameterization/SGD batch size).
> 2) We derive a novel closed form expression for the “noise scale” of the SGD, which holds even when the covariance matrix between the gradients of different parameters is non-stationary. We exploit this expression to predict three scaling rules between the batch size, learning rate, training set size and momentum coefficient.
> 3) We verified these scaling rules empirically, and we believe they will prove extremely useful to ML practitioners. We note that while the B ~ a/(1-m) scaling rules enable us to achieve large batch training without hyper-parameter tuning, the B ~ N rule is equally valuable; since it enables us to retrain production models on new training data without retuning the batch size.
>
> With current tools, the Gaussian approximation to the evidence cannot be estimated efficiently in deep networks. However this is not in fact a major limitation, since in practical scenarios we will always rely on test sets. What is important is to build an intuition for the factors which control generalization, and to use this intuition to improve the accuracy of our models. This is why we presented generalization and the SGD in a single paper; the intuition we gained from the approximate Bayesian evidence resolves the sharp minima debate, and the trade off between the depth and breadth of minima in the evidence explains how we should tune SGD hyper-parameters to maximize the test set accuracy.
>
> Unfortunately, one cannot derive a stationary distribution for the SGD in the infinite time limit unless one imposes the unrealistic assumption that the gradient covariances are stationary. As a result, it is very challenging to formally prove that SGD converges to minima with large evidence. However our argument runs as follows: if one were to assume the covariances were stationary, one could derive a stationary distribution for the SGD in the infinite limit [1] and one could formally prove that SGD converges to models of large evidence at an optimal noise scale. While we cannot formally prove this for the general case, we expect noise to have similar effects, and this matches what we observe empirically.
>
> To respond to the remaining comments:
> 1) We will add appropriate citations/discussion to the text.
> 2) We have explored the average margin of models trained on random and informative labels. We find that the margin is ~50% larger when trained on random labels in our experiments. However unlike the evidence, the margin is not strongly correlated with either the test cross-entropy or the test accuracy.
> 3/4) See above. More specifically, the SGD would generate Bayesian posterior samples if the covariance matrix were isotropic and stationary. In this case one can prove formally that there is an optimal noise scale which biases SGD towards minima with large evidence. Our observation is that this optimal noise scale persists empirically even though the true covariance matrix is anisotropic and non-stationary. We will discuss this connection in more depth when we update the manuscript.
> 5) Yes, if we kept aN/B constant then the test set accuracy would be constant and there would not be a generalization gap (until the learning rate a is too large, as seen in figure 5a); this is a key result of the paper and the meaning of the scaling rules. We will edit the text to make this clearer. The optimal batch size only arises when one holds the learning rate constant, equivalently there would be an optimal learning rate at constant batch size. We used SGD with momentum since it is more popular than conventional SGD in practice. The results without momentum are the same (both optimal noise scale and the scaling rules).
> 6) The derivation in section 2 was first performed by Mackay in 1992. We include it here because many researchers have not seen it, and because it is central to the remainder of the paper. In addition, we demonstrate the Bayesian evidence penalizes sharp minima but is invariant to model parameterization, resolving the objections of [2]. The derivation in section 5 is original. While our treatment is similar to [1], they assume that the covariance matrix is stationary; we show that this assumption is not necessary, since the covariance matrix cancels out when one equates the SGD to the SDE.
> 7) We apologize for this. We normalized the expected length of the training examples, not the features.
>
> [1] Mandt et al., Stochastic Gradient Descent as Approximate Bayesian Inference, ICML 2017
> [2] Dinh et al., Sharp minima can generalize for deep nets, ICML 2017

---

### Official Review · AnonReviewer2 · 2017-11-28
**Useful insights**

**Rating:** 7
**Confidence:** 3

**Review:**

This paper builds on Zhang et al. (2016) (Understanding deep learning requires rethinking generalization). Firstly, it shows experimentally that the same effects appear even for simple models such as linear regression.  It also shows that the phenomenon that sharp minima lead to worse result can be explained by Bayesian evidence.  Secondly, it views SGD with different settings as introducing different levels of noises that favors different minima. With both theoretical and experimental analysis, it suggests the optimal batch-size given learning rate and training data size. The paper is well written and provides excellent insights.

Pros:
1. Very well written paper with good theoretical and experimental analysis.
2. It provides useful insights of model behaviors which are attractive to a large group of people in the community.
3. The result of optimal batch size setting is useful to wide range of learning methods.

Cons and mainly questions:
1. Missing related work.
One important contribution of the paper is about optimal batch sizes, but related work in this direction is not discussed. There are many related works concerning adaptive batch sizes, such as [1] (a summary in section 3.2 of [2]).

2. It will be great if the author could provide some discussions with respect to the analysis of information bottleneck [3] which also discuss the generalization ability of the model.

3. The result of the optimal mini-batch size depends on the training data size. How about real online learning with streaming data where the total number of data points are unknown?

4. The results are reported mostly concerning the training iterations, not the CPU time such as in figure 3. It will be fair/interesting to see the result for CPU time where small batch maybe favored more.


[1] Balles, Lukas, Javier Romero, and Philipp Hennig. "Coupling Adaptive Batch Sizes with Learning Rates." arXiv preprint arXiv:1612.05086 (2016).
[2] Zhang, Cheng, Judith Butepage, Hedvig Kjellstrom, and Stephan Mandt. "Advances in Variational Inference." arXiv preprint arXiv:1711.05597 (2017).
[3] Tishby, Naftali, and Noga Zaslavsky. "Deep learning and the information bottleneck principle." In Information Theory Workshop (ITW), 2015 IEEE, pp. 1-5. IEEE, 2015.

—————-
Update: I lowered my rating considering other ppl s review and comments.

---

> ### Public Comment · (anonymous) · 2017-12-03
> **I wonder does the reviewer know anything about bayesian optimization**
>
> What the paper said is a common sense in  the bayesian opimization community!
>
> I can find tons of papers revealing the same idea!
>
> Their strength is only to show the noise level is associated with batchsize, which is a naive idea I think everyone konws

---

> > ### Public Comment · (anonymous) · 2017-12-03
> > **The tons of papers**
> >
> > C. Chen, D. Carlson, Z. Gan, C. Li, and L. Carin. Bridging the gap between stochastic gradient MCMC and stochastic optimization. In AISTATS, 2016.
> >
> > T. Chen, E. B. Fox, and C. Guestrin. Stochastic gradient Hamiltonian Monte Carlo. In Proceedings of the 31st International Conference on Machine Learning, pages 1683–1691, 2014.
> >
> > M. Welling and Y. W. Teh. Bayesian learning via stochastic gradient langevin dynamics. In ICML, 2011.
> >
> >
> > https://arxiv.org/pdf/1707.05947

---

> > > ### Public Comment · (anonymous) · 2017-12-03
> > > **Supplementary instructions**
> > >
> > > They all reveal the idea that bayesian methods will goes to the flat minima which will void over-fitting , even some of them have prove the generalization bound in non asymptotic time!

---

> > ### Comment · AnonReviewer2 · 2017-12-03
> > **The reviewer does know BO and still vote to accept the paper**
> >
> > Yes. I do know Bayesian optimization and I also pointed out that this paper lacks discussion of related work in my comment and added an example of the paper [1] from P. Hennig (who did a lot of work in BO and the paper that I referred to was one exemplar paper that propose the optima batchsize given learning rate. ).
> >
> > Thank you  to add more related worked that the author of the paper  should discuss.
> >
> > I agree with you the these idea are not completely novel and the paper lacks discussion about related work.
> >
> > However, I do also recognize the contribution of the paper. Although they are not the first one that propose  analysis of different minia and the analysis of optimal batchsize,  the setting and analysis differs from existing work, and I think that such idea should be discussed more in deep learning community. I thus keep my vote on accepting the paper.

---

> > > ### Public Comment · (anonymous) · 2017-12-04
> > > **-**
> > >
> > > This comment has been redacted for violating the ICLR 2018 anonymity policy.

---

> > > ### Public Comment · (anonymous) · 2017-12-04
> > > **-**
> > >
> > > The idea of using SDE is also well-known
> > > Mandt S, Hoffman M D, Blei D M. Stochastic Gradient Descent as Approximate Bayesian Inference. In ICML2017.
> > > Pratik Chaudhari, Stefano Soatto Stochastic gradient descent performs variational inference, converges to limit cycles for deep networks arXiv preprint

---

> > > ### Public Comment · (anonymous) · 2017-12-04
> > > **-**
> > >
> > > Best of my knowledge, the first one to use SDE to analysis sgd is
> > >
> > > Qianxiao Li, Cheng Tai, Weinan E Stochastic modified equations and adaptive stochastic gradient algorithms ICML2017(arxiv preprint:2015)

---

> ### Author Response · Authors · 2017-12-06
> **Response to anonymous comments**
>
> We would like to respond to the comments from anonymous commenter(s) which followed your review. We feel that these comments are misleading and misrepresent the contributions of this work; which we believe are significant.
>
> We have reviewed the previous papers suggested by the commenter(s). Two are already cited in our paper [1,2]. One was released on arXiv after we submitted [3]. We are happy to cite [4] when we update the manuscript, which proposes merging SGD with control theory. The other suggested papers [5-7] are not relevant, since they discuss posterior sampling methods which use stochastic gradients, not the SGD.
>
> We emphasize that none of these papers predicted or observed an optimal batch size (at constant learning rate). Additionally we are the first to derive the three scaling rules. Our treatment of the SGD is most similar to [2], however their analysis only holds near local minima where the gradient covariance matrix between parameters is stationary. By contrast our derivation applies throughout training for both stationary and non-stationary covariance matrices. This is important since [8] found that the benefits of noise are most pronounced at the start of training. Furthermore, our results have important practical applications:
>
> i) We show that tuning the batch size can lead to surprisingly large improvements in test accuracy.
> ii) The scaling rules enable us to achieve large batch training without reducing the test set accuracy and without additional hyper-parameter tuning.
> iii) They also predict how the batch size should be changed over time as more training data is collected (the most common reason to retrain a model is that one has collected more training data).
> iv) Finally, the scaling rules enable us to compare training runs performed on different hardware with different batch sizes/learning rates/momentum coefficients.
>
> We have submitted two papers to ICLR this year. The second paper is an extension of this work, and we make this clear in the introduction of the second paper. It is up to the reviewers of this second paper to judge if the extension is sufficient to merit acceptance.
>
> [1] M. Welling and Y. W. Teh., Bayesian learning via stochastic gradient langevin dynamics, In ICML 2011.
> [2] Mandt S et al., Stochastic Gradient Descent as Approximate Bayesian Inference, In ICML 2017.
> [3] Pratik Chaudhari and Stefano Soatto, Stochastic gradient descent performs variational inference, converges to limit cycles for deep networks, arXiv preprint
> [4] Qianxiao Li et al., Stochastic modified equations and adaptive stochastic gradient algorithms,  ICML 2017 (arxiv 2015)
> [5] C. Chen et al., Bridging the gap between stochastic gradient MCMC and stochastic optimization, In AISTATS 2016.
> [6] T. Chen et al., Stochastic gradient Hamiltonian Monte Carlo, In ICML 2014
> [7] W. Mou et al., Generalization Bounds of SGLD for Non-convex Learning: Two Theoretical Viewpoints, arXiv preprint
> [8] Keskar et al., On large-batch training for deep learning: Generalization gap and sharp minima, ICLR 2017

---

> ### Author Response · Authors · 2017-12-06
> **Response to official review**
>
> We thank the referee again for their positive assessment of our work. In response to the comments and questions raised in the official review:
>
> 1) We did not cite works on adaptive batch sizes, since we considered the simpler case of constant SGD (constant learning rate/batch size). However the reviewer is correct that these works are relevant to the optimal batch size discussion, particularly “Coupling Adaptive Batch Sizes with Learning Rates” which also proposes a linear scaling rule. We apologize for this and will cite these works when we update the manuscript.
>
> 2) Yes, we believe that there is an extremely close relationship between Bayesian model comparison and some aspects of the information bottleneck principle. For instance, Bayesian model comparison essentially minimizes the information content of the parameters (the “minimum description length” principle), which is conceptually very close to minimizing the mutual information content between the intermediate representations and the inputs. However before we say more, we would like to study the information bottleneck more closely.
>
> 3) This is a very interesting question. In conventional online learning, one only uses each training example once. This is different to the Bayesian perspective, where one should re-use all old samples repeatedly. Consequently I’m not sure there is a principled answer. However intuitively, it would be sensible to increase the batch size or decay the learning rate proportional to the total number of training examples seen to date, thus ensuring that the noise scale is constant as the training set grows.
>
> 4) We intentionally ran most of our comparisons at constant number of training iterations, to ensure that the total “time” simulated by the underlying SDE of each curve was the same (equivalently ensuring that the expected distance travelled from the initialization point was the same in all cases). We believe that this provides a more meaningful comparison than holding the CPU time constant. However we are happy to add additional experiments to the appendix.
>
> Best wishes

---

### Public Comment · (anonymous) · 2017-11-05
**Question on memorizing under linear models**

It is stated in the paper that linear models have the behavior as deep nets that generalize well on informative labels but can also memorize random labels of the same inputs. Is this just because in the synthetic experiment the number of training instance is 200, which is strictly less than the number of model parameters, which is 201 = 200 + 1? Since the model is linear, this amounts to solving an underdetermined system of linear equations, which is guaranteed to have a solution with probability 1 (assuming the inputs are randomly sampled, which is the case in the experiment). I am wondering whether the same phenomenon can be observed when the training set size is larger? Say, 500? Thanks!

---

> ### Author Response · Authors · 2017-11-08
> **Exactly!**
>
> Thank you for your interest in our work!
>
> Yes, that is exactly why the linear model can memorize random labels of the training data; it is "over-parameterized". A typical rule of thumb is linear models can memorize roughly two labels per parameter. This is also exactly why the deep networks in Zhang et al. can memorize random labels of their training data; they also have more parameters than training examples. In both linear and deep networks, if we made the training set sufficiently large we wouldn't be able to memorize randomly labelled data. I did a quick check and our model can memorize about 300 labels, it can't memorize 500.
>
> What surprised Zhang et al. is that the model did generalize well to the test set when trained on real informative labels, even though they had shown that the model was sufficiently flexible to assign random labels to the same inputs (ie meaningless solutions which do not generalize do exist). Again, we show exactly the same phenomenon here. The purpose of showing these results in a linear model is precisely to make these results intuitive, and to demonstrate that they should not be explained by any property unique to deep learning.
>
> We show that good minima which generalize well to the test set can be distinguished from bad minima which do not by evaluating the Bayesian evidence (also known as the marginal likelihood). This evidence is a weighted combination of the depth and breadth of a minimum, but it is invariant to model parameterization, which clarifies the recent debate surrounding sharp minima.

---

### Author Response · Authors · 2017-12-22
**Updated manuscript**

We have uploaded an updated version of the manuscript, which we believe significantly strengthens the paper. As well as fixing some minor issues raised by the reviewers, the main changes are:

1) To respond to the comments of reviewer 3, we have replaced the synthetic data experiments in section 3 with real data experiments on MNIST. The conclusions of the section are unchanged; we observe the exact same phenomenon observed by Zhang et al. in a linear model, and show that this phenomenon can be understood via the Bayesian evidence. We also present a brief discussion of the training set margin when trained on random/informative labels, as requested by reviewer 1.

2) We have introduced a new section to the appendix discussing Bayesian posterior sampling, which provides a simple case where noise provably drives the parameters towards broad minima whose evidence is large. We have edited section 4 to emphasize the connection between these results and SGD more clearly. We also added another section to the appendix briefly discussing the Gaussian approximation to minibatch gradient noise.

3) We edited the introduction to highlight our contributions more clearly. We have also highlighted that the claims of Zhang et al. regarding learning theory are disputed. These claims are not important to our work, which understands the memorization phenomenon Zhang et al. presented and resolves the debate surrounding sharp minima and model parameterization.

4) We have included a brief discussion of the stochastic differential equation discretization error in section 5. We also emphasize the potential practical applications of the training set size scaling rule. We clarify the novelty of our treatment compared to previous works.

5) We have added additional citations to the text. However we found that many of the papers posted anonymously following reviewer 2’s highly positive initial score (9) were not relevant to our work, as they studied specific bayesian posterior sampling methods, whereas our work provides bayesian insights on SGD itself. We discuss this further in the comments below.

We would also like to comment that Reviewer 3’s primary criticism of our work is “the paper is very much focussed” on Zhang et al.’s rethinking generalization paper. This is simply not the case; our work makes a number of novel contributions. Indeed the bulk of the text is devoted to our discussion of SGD, the optimal batch size and the scaling rules. Many papers have responded to Zhang et al.’s findings, and reviewers 1 and 2 both felt we made an important contribution to this debate.

---

### Decision · Program_Chairs · 2018-01-29
**ICLR 2018 Conference Acceptance Decision**

**Decision:**

Accept (Poster)

**Comment:**

I'm inclined to recommend accepting this paper, although it is borderline given the strong dissenting opinion. The revisions have addressed many of the concerns about quality, clarity, and significance. The paper gives an end to end explanation in Bayesian terms of generalization in neural networks using SGD.

However, it is my opinion that Bayesian statistics is not, at present, a theory that can be used to explain why a learning algorithm works. The Bayesian theory is too optimistic: you introduce a prior and model and then trust both implicitly. Relative to any particular prior and model (likelihood), the Bayesian posterior is the optimal summary of the data, but if either part is misspecified, then the Bayesian posterior carries no optimality guarantee. The prior is chosen for convenience here. And the model (a neural network feeding into cross entropy) is clearly misspecified.

However, there are ways to sidestep both these issues using a frequentist theory closely related to Bayes, which can explain generalization. Indeed, you cite a recent such paper by Dzugate and Roy who use PAC-Bayes. However, you citation is disappointingly misleading: a reader would never know that these authors are also responding to Zhang, have already proposed to explain "broad minima" in (PAC-)Bayesian terms, and then even get nonvacuous bounds. (The connection between PAC-bayes and marginl likelihood is explained by Germain et al. "PAC-Bayesian Theory Meets Bayesian Inference").  Dzugate et al don't propose to explain why SGD finds such "good" minima. So I would say, your work provides the missing half of their argument. This work deserves more prominent placement and shouldn't be buried on page 5. Indeed, it should appear in the introduction and a proper description of the relationship should be given.